# Tolerance interval testing for assessing accuracy and precision simultaneously

**Chieh Chiang, Chin-Fu Hsiao***

Institute of Population Health Sciences, National Health Research Institutes, Zhunan, Taiwan

* chinfu@nhri.org.tw

## Abstract

Tolerance intervals have been recommended for simultaneously validating both the accuracy and precision of an analytical procedure. However, statistical inferences for the corresponding hypothesis testing are scarce. The aim of this study is to establish a whole statistical inference for tolerance interval testing, including sample size determination, power analysis, and calculation of p-value. More specifically, the proposed method considers the bounds of a tolerance interval as random variables so that a bivariate distribution can be derived. Simulations confirm the theoretical properties of the method. Furthermore, an example is used to illustrate the proposed method.

**Data Availability Statement:** All relevant data are within the paper.

**Funding:** The author(s) received no specific funding for this work.

## 1. Introduction

When assessing whether an analytical procedure is suitable for its intended purpose, the impacts of accuracy and precision are usually considered. "Accuracy" usually refers to the expectation of the effect response from the product, whereas "precision" is the variability of the effect response from the product. In practice, the two parameters are unknown and need to be estimated. If the two parameters are validated separately, then multiple adjustments of the controls of family-wise error rates for making the wrong decision are necessary. However, an analytical procedure usually allows for a product to have a relatively small value of variation, accommodating a relatively greater value of bias than a product with a greater value of variation. For these reasons, the United States Pharmacopeia (USP) guideline <1210> *Statistical Tools for Procedure Validation* [1] recommends a two-sided tolerance interval as being useful for establishing a single criterion to simultaneously validate both accuracy and precision; in other words, an assessment is useful when assessing whether $100\gamma$ percent of a population, say $X$, is located within a prespecified acceptable interval $(c_L, c_U)$.

Tolerance interval approaches have been widely used in the area of sampling acceptance criteria, however, the relationship between hypothesis testing and tolerance interval sampling acceptance plan were scarcely discussed [3]. A hypothesis testing for assessing the drug effect is usually required, and thus controlling the type I error rate and achieving the desired power are important. Therefore, Novick et al. [2] and Dong et al. [3] suggested two one-sided tolerance interval tests to dose content uniformity tests, delivered dose uniformity tests, and dissolution tests. In their applications, the hypotheses $H_{0L}:\Pr(X<c_L)\geq P_1$ and $H_{0U}:\Pr(X<c_U)\geq P_2$

**Competing interests:** The authors have declared that no competing interests exist.

were tested, respectively, where $X$ is a random variable from a population with prespecified constants $c_L$, $c_U$, $P_1$, and $P_2$. However, Novick et al. had pointed out that the use of two one-sided tolerance intervals is correct for controlling of the type I error rate only if the variability of the population is sufficient small. If so, the use of the tolerance interval test seems to be meaninglessness since it is essentially equivalent to testing merely the population mean. Moreover, whether the variability of the population is small enough is usually unknown in practice.

In this study, a two-sided tolerance interval test is considered. As pointed out by Chiang et al. [4], there must be two unknown parameters $\theta_L$ and $\theta_U$ leading to $P(\theta_L<X<\theta_U)\geq\gamma$. Therefore, when being linked to the prespecified acceptable interval, one of the following four situations must be true: (i) $c_L\leq\theta_L$ and $\theta_U\leq c_U$, which is what we expect; (ii) $\theta_L\leq c_L$ and $\theta_U\leq c_U$, which must indicate that the expectation of $X$ has a negative bias from our expectation; (iii) $c_L\leq\theta_L$ and $c_U\leq\theta_U$, which must indicate that the expectation of $X$ has a positive bias from our expectation; and (iv) $\theta_L<c_L$ and $c_U<\theta_U$, which must indicate that the variability of $X$ exceeds what we expected. Consequently, situations (ii) and (iii) represent a lack of accuracy, whereas situation (iv) represents a lack of precision; these three situations should be rejected. Therefore, it is indicated that the statistical hypotheses for testing $\theta_L$ and $\theta_U$ are as follows:

$$H_0 : \theta_L < c_L \text{ or } \theta_U > c_U \text{ versus } H_a : c_L \leq \theta_L \text{ and } \theta_U \leq c_U. \tag{1}$$

For these hypotheses, the accuracy and precision can be assessed simultaneously in a single test without multiple adjustments of the type I error rates.

The corresponding test statistic for hypotheses (1) is exactly a two-sided tolerance interval because, by definition, a $100(1-\alpha)\%$ confidence $100\gamma\%$ content two-sided tolerance interval of $X$ satisfies the following equation:

$$P_{L,U}[P_X(L < X < U|L, U) \geq \gamma] = 1 - \alpha, \tag{2}$$

where $L$ and $U$ are called the lower and upper tolerance limits, respectively. A general introduction and discussion of tolerance intervals can be found in the book by Krishnamoorthy and Mathew [5]. Naturally, from Eq (2), $L$ and $U$ are estimators of $\theta_L$ and $\theta_U$, respectively, with a probability of $1-\alpha$. This indicates that $P(L\leq\theta_L \text{ or } \theta_U\leq U) = 1-\alpha$. Therefore, when the null hypothesis is true, $P(c_L<L \text{ and } U\leq c_U) = \alpha$ controls the type I error rate at $\alpha$. Consequently, statistical quality is declared with significance level $\alpha$ if $l>c_L$ and $u<c_U$, wherein $l$ and $u$ are observations of $L$ and $U$, respectively.

On the other hand, the sample size determination for a tolerance interval is traditionally used to achieve a desired width for the interval [6, 7]. In doing so, only the control of precision is considered in the traditional sample size determination for a tolerance interval. Now, if hypothesis testing and the tolerance interval sampling acceptance plan are linked, determining sample size for a desired power of the tolerance interval test is equivalent to providing a sufficiently large probability of the tolerance interval falling within a prespecified acceptance interval; that is, accuracy and precision are taken into consideration simultaneously in the sample size determination. Therefore, evaluating the required sample size for a two-sided tolerance interval test is also an important aim of this study.

The rest of this paper is arranged as follows. In Section 2, the tolerance interval proposed by Howe [8] and recommended by the USP guideline is described. Then, a power function is derived from the asymptotic distribution for the lower and upper tolerance limits. The sample size can then be set to reach the required level of power. The p-value of the tolerance interval test is derived by a similar procedure. In Section 3, the proposed method is illustrated by an example drawn from the USP guideline. The good properties of the method are confirmed by

simulations in Section 4. We study the required sample size as a function of the parameters on sample size in this section. The last section provides final remarks and discussion.

## 2. Tolerance interval testing

### 2.1. Statistical assumption and interval estimation

Let $X_i$ be the reportable response for $i = 1,\ldots,n$. Suppose that these responses are independent and identically distributed normal variables such that

$$X_i \sim N(\mu, \sigma^2) \text{ for } i = 1, \ldots, n, \tag{3}$$

where $N(\mu, \sigma^2)$ is the normal distribution with mean $\mu$ and variance $\sigma^2$. Denote the sample means and sample variances, respectively, as $\overline{X} = \sum_{i=1}^{n} X_i/n$ and $S^2 = \sum_{i=1}^{n} (X_i - \overline{X})^2/(n-1)$. A two-sided $100(1-\alpha)\%$ confidence $100\gamma\%$ content tolerance interval can then be constructed as follows:

$$[L, U] = [\overline{X} \pm kS]. \tag{4}$$

Here $k$, which represents the tolerance factor, does not have a closed-form solution and must be evaluated by numerical methods. The exact tabulated values of $k$ can be found in [9]. However, an approximation suggested by Howe [8] works well in practical situations if exact values are not available and, therefore, is used in the USP guideline [1] as follows:

$$k = \sqrt{\frac{z_{(1+\gamma)/2}^2 (n-1)}{\chi_{\alpha,n-1}^2} \left(1 + \frac{1}{n}\right)}, \tag{5}$$

where $z_{(1+\gamma)/2}$ is a standard normal percentile with area $(1+\gamma)/2$ to the left and $\chi_{\alpha,n-1}^2$ is a chi-squared percentile with area $\alpha$ to the left and $n-1$ degrees of freedom. Consequently, the accuracy and precision can be validated simultaneously with a significance level $\alpha$ if $[L, U]$ is contained in $(c_L, c_U)$.

Obviously, appropriately setting the acceptable limits $c_L$ and $c_U$ is the key point for the correct assessment of accuracy and precision. In doing so, $c_L$ and $c_U$ are recommended to be at least the expected values of $\mu \pm 3\sigma$ since it is well-known that 99.73% of $X$ is included within this range. If $\gamma$ is not large, for example, 90%, then the acceptance limits may be changed to $\mu \pm 2\sigma$; that is, 95.45% of $X$ should be included.

### 2.2. Sample size determination

According to the rejection rule of the two-sided tolerance interval testing, the power function is written as

$$\pi(\boldsymbol{\theta}) = P\{[L, U] \subset (c_L, c_U) | \boldsymbol{\theta}\}, \tag{6}$$

where $\boldsymbol{\theta}$ denotes a vector of parameters $\mu$ and $\sigma$. Since the lower and upper bounds themselves are random variables, Eq (6) can be rewritten as

$$\pi(\boldsymbol{\theta}) = P\{L > c_L, U < c_U | \boldsymbol{\theta}\}. \tag{7}$$

To calculate this probability, we need to find the joint distribution of $L$ and $U$. From (3), $L$ and $U$ are represented as the combination of the sample mean $\overline{X}$ and length $kS$. It is clear that $\overline{X}$ follows a normal distribution with mean $\mu$ and variance $\sigma^2/n$. Also, since $(n-1)S^2/\sigma^2$ follows a chi-square distribution with degrees of freedom $n-1$, we have that $\sqrt{n-1}S/\sigma$ follows a chi

distribution with degrees of freedom $n-1$. This implies that the sample standard deviation $S$ converges to a normal random variable with mean

$$\mu_S = \sigma\sqrt{2/(n-1)}\Gamma(n/2)/\Gamma((n-1)/2)$$

and variance

$$\sigma_S^2 = \sigma^2\{1 - 2\Gamma^2(n/2)/\Gamma^2((n-1)/2)\}.$$

Here $\Gamma$ is the gamma function. Consequently, $L = \overline{X} - kS$ and $U = \overline{X} + kS$ follow a bivariate normal distribution asymptotically with a mean vector $[\mu-k\mu_S, \mu+k\mu_S]$' and covariance matrix

$$\begin{bmatrix} \sigma^2/n + k^2\sigma_S^2 & \sigma^2/n - k^2\sigma_S^2 \\ \sigma^2/n - k^2\sigma_S^2 & \sigma^2/n + k^2\sigma_S^2 \end{bmatrix}.$$

More details for the derivations of the asymptotic bivariate normal distribution are provided in S1 Appendix. Based on the above asymptotic distribution, the probability in (6) can be re-expressed as

$$\pi(\boldsymbol{\theta}) = F_\rho\left(\frac{-\tau_L + \mu - k\mu_S}{\sqrt{\sigma^2/n + k^2\sigma_S^2}}, \frac{\tau_U - \mu - k\mu_S}{\sqrt{\sigma^2/n + k^2\sigma_S^2}} \Big| \boldsymbol{\theta}\right), \tag{8}$$

where $F_\rho(z_{-L}, z_U|\boldsymbol{\theta})$ denotes the cumulative distribution function of the standard bivariate normal distribution for the standardized random variables $-L$ and $U$ with the following correlation:

$$\rho = (-\sigma^2/n + k^2\sigma_S^2)/(\sigma^2/n + k^2\sigma_S^2).$$

For a pair of parameters, the required sample size is determined by insisting that the power exceeds a set value. S2 Appendix provides an SAS code for sample size determination that is based on the SAS nonlinear problem (NLP) procedure. This SAS code allows users to specify the design parameters of the content level, confidence level, desired level of power, alternative mean and variance, and accepted reference values.

Since the asymptotic distribution of the lower and upper tolerance bounds has been derived, it can be applied to calculate the p-value of the tolerance interval test. Specifically, given observations of $L$ and $U$ –say $l$ and $u$, respectively–the p-value is

$$p-\text{value} = \max_{\boldsymbol{\theta}\in H_0} F_\rho\left(\frac{-l + \mu - k\mu_S}{\sqrt{\sigma^2/n + k^2\sigma_S^2}}, \frac{u - \mu - k\mu_S}{\sqrt{\sigma^2/n + k^2\sigma_S^2}} \Big| \boldsymbol{\theta}\right). \tag{9}$$

Note that, under the normality assumption, there are infinite sets of means and standard deviations satisfying the null hypothesis (1). Hence, a Lagrange multiplier method is used to evaluate the maximum $p$-value; we therefore provide another SAS code in S3 Appendix.

## 3. Example

The example of high-performance liquid chromatography mentioned in the USP document [1] is used to illustrate the proposed study. The unit of measurement for each reportable value is the mass fraction of drug substance expressed in units of mg/g and does not change as the level of concentration varies. The sample mean and sample standard deviation are 992.81 and 4.44, respectively, with a sample size of 9. For a content level of 90% and a confidence level of

90%, the Howe approximation of $k$ is

$$k = \sqrt{\frac{1.64^2(9-1)}{3.49}\left(1+\frac{1}{9}\right)} = 2.63.$$

It follows that the 90% confidence, 90% content tolerance interval is

$$[992.81 \pm 2.63 \times 4.44] = [981.2, 1004.5].$$

Suppose the criterion is designed to ensure that the difference between a reference accuracy of 1,000 and the acceptable limits are less than 2%; specifically, the tolerance interval falls between 980 and 1,020. In this example, it is obvious that accuracy and precision are both validated. In addition, the p-value is 0.0218, which is much less than the nominal level of 10%.

If the mean and standard deviation are used to design a new test with the same acceptable range, the proposed sample size determination indicates that, for a content level of 90% and a confidence level of 90%, merely 4 subjects are required to meet a power of 80%. In fact, the theoretical power, via the use of the proposed method, is 92.96%. If the acceptable range is reduced to [990, 1010], the proposed sample size determination indicates that for a content level of 90% and a confidence level of 90%, 43 subjects are required to meet a power of 80%. More specifically, when the sample size is 43, the lower and upper tolerance limits follow a bivariate normal distribution asymptotically with the mean vector [998.58, 1001,42]' and covariance matrix

$$\begin{bmatrix} 19.71 & -6.23 \\ -6.23 & 19.71 \end{bmatrix}.$$

This results in a power of 0.8059 for the tolerance interval test.

## 4. Simulation and numerical study

The purpose of this simulation is to investigate whether the proposed sample size determination can reach the targeted level of statistical power under several combinations of design parameters. As in the USP example, we set, without loss of generality, $\mu$ from 0 to 1 in increments of 0.5 and $\sigma$ from 3 to 4 in increments of 0.5. The acceptable region $(c_L, c_U) = (-c, c)$ with $c$ ranging from 10 to 12 in increments of 1. The confidence level and content level are $\alpha = 0.1$ and $\gamma = 0.1$, respectively. Consequently, there are 27 sets of parameters for the simulation. One million random samples of a size determined by the proposed method are generated from the normality assumption in (1) for each set ofparameter components. The empirical power is the proportion of the 1,000,000 two-sided tolerance intervals that are contained in the criterion $(-c, c)$. The coverage probability is simultaneously verified, and the empirical result is the proportion of the 1,000,000 lower and upper tolerance limits, say $l^*$ and $u^*$, satisfying $F(u^*) - F(l^*) > 90\%$, where $F(.)$ is the marginal cumulative distribution function of (1).

The simulation results are presented in Table 1. There are several points we wish to make. First, for the 27 different sets of parameters, all of the empirical powers are greater than the desired level of 80%, which demonstrates that the proposed sample size determination can provide sufficient power under various sets of parameters for validating both accuracy and precision simultaneously based on the two-sided tolerance interval. Moreover, the asymptotic and empirical powers are quite consistent since all of the absolute differences between the two values are less than or equal to 0.0027. In addition, the simulation study shows that the resultant power is stable even when the sample size is very small. For example, the minimum sample size is 7 for $\mu = 0$, $\sigma = 3$, and $c = 12$; the difference between the asymptotic and empirical powers is merely -0.0027. Finally, the empirical coverage probabilities are approximately 90%.

**Table 1. Sample size and quantile determination at a confidence level of 90%, a content level of 90%, and a desired power of 80%.**

| μ | σ | c | Total sample size | Coverage Probability | Asymptotic | Power Empirical | Difference |
|---|---|---|---|---|---|---|---|
| 0.0 | 3.0 | 10 | 10 | 0.8974 | 0.8401 | 0.8391 | -0.0010 |
| | | 11 | 8 | 0.8982 | 0.8377 | 0.8369 | -0.0008 |
| | | 12 | 7 | 0.8982 | 0.8592 | 0.8565 | -0.0027 |
| 0.0 | 3.5 | 10 | 15 | 0.8975 | 0.8196 | 0.8201 | 0.0005 |
| | | 11 | 11 | 0.8978 | 0.8090 | 0.8100 | 0.0010 |
| | | 12 | 9 | 0.8972 | 0.8200 | 0.8204 | 0.0003 |
| 0.0 | 4.0 | 10 | 25 | 0.8982 | 0.8133 | 0.8141 | 0.0008 |
| | | 11 | 17 | 0.8976 | 0.8155 | 0.8158 | 0.0003 |
| | | 12 | 13 | 0.8972 | 0.8259 | 0.8267 | 0.0007 |
| 0.5 | 3.0 | 10 | 10 | 0.8972 | 0.8236 | 0.8241 | 0.0004 |
| | | 11 | 8 | 0.8976 | 0.8255 | 0.8254 | -0.0001 |
| | | 12 | 7 | 0.8988 | 0.8500 | 0.8485 | -0.0014 |
| 0.5 | 3.5 | 10 | 16 | 0.8968 | 0.8281 | 0.8284 | 0.0003 |
| | | 11 | 12 | 0.8973 | 0.8383 | 0.8372 | -0.0011 |
| | | 12 | 9 | 0.8974 | 0.8089 | 0.8101 | 0.0012 |
| 0.5 | 4.0 | 10 | 27 | 0.8979 | 0.8151 | 0.8161 | 0.0009 |
| | | 11 | 18 | 0.8967 | 0.8222 | 0.8220 | -0.0001 |
| | | 12 | 13 | 0.8970 | 0.8122 | 0.8129 | 0.0007 |
| 1.0 | 3.0 | 10 | 11 | 0.8979 | 0.8243 | 0.8243 | -0.0001 |
| | | 11 | 9 | 0.8978 | 0.8530 | 0.8508 | -0.0021 |
| | | 12 | 7 | 0.8982 | 0.8231 | 0.8233 | 0.0001 |
| 1.0 | 3.5 | 10 | 18 | 0.8973 | 0.8175 | 0.8182 | 0.0007 |
| | | 11 | 13 | 0.8972 | 0.8327 | 0.8322 | -0.0005 |
| | | 12 | 10 | 0.8976 | 0.8324 | 0.8325 | 0.0001 |
| 1.0 | 4.0 | 10 | 33 | 0.8977 | 0.8050 | 0.8059 | 0.0010 |
| | | 11 | 20 | 0.8975 | 0.8111 | 0.8116 | 0.0005 |
| | | 12 | 14 | 0.8965 | 0.8081 | 0.8091 | 0.0010 |

Next, the impacts of the magnitudes of the mean, standard deviation, and criterion on sample size determination are explored in Fig 1. The figure demonstrates that the required sample size increases as the mean and standard deviation increase and decreases as the criterion increases. Note that here, a non-zero mean indicates a bias of accuracy. The relation between the sample size and parameters is, therefore, intuitively correct because the increases in bias and variability must increase the number of samples required to achieve the targeted level of power. On the other hand, the increase in the acceptable margin facilitates the validation of accuracy and precision; hence, the required sample size decreases.

## 5. Discussion and final remarks

Tolerance intervals have been recommended, for example, by the abovementioned USP document, to simultaneously assess accuracy and precision. This study provides a connection between two-sided hypothesis testing and a two-sided tolerance interval-based assessment. Simulations show that the proposed approach provides sufficient and consistent results compared with the theoretical values on various combinations of parameters even when the sample size is small.

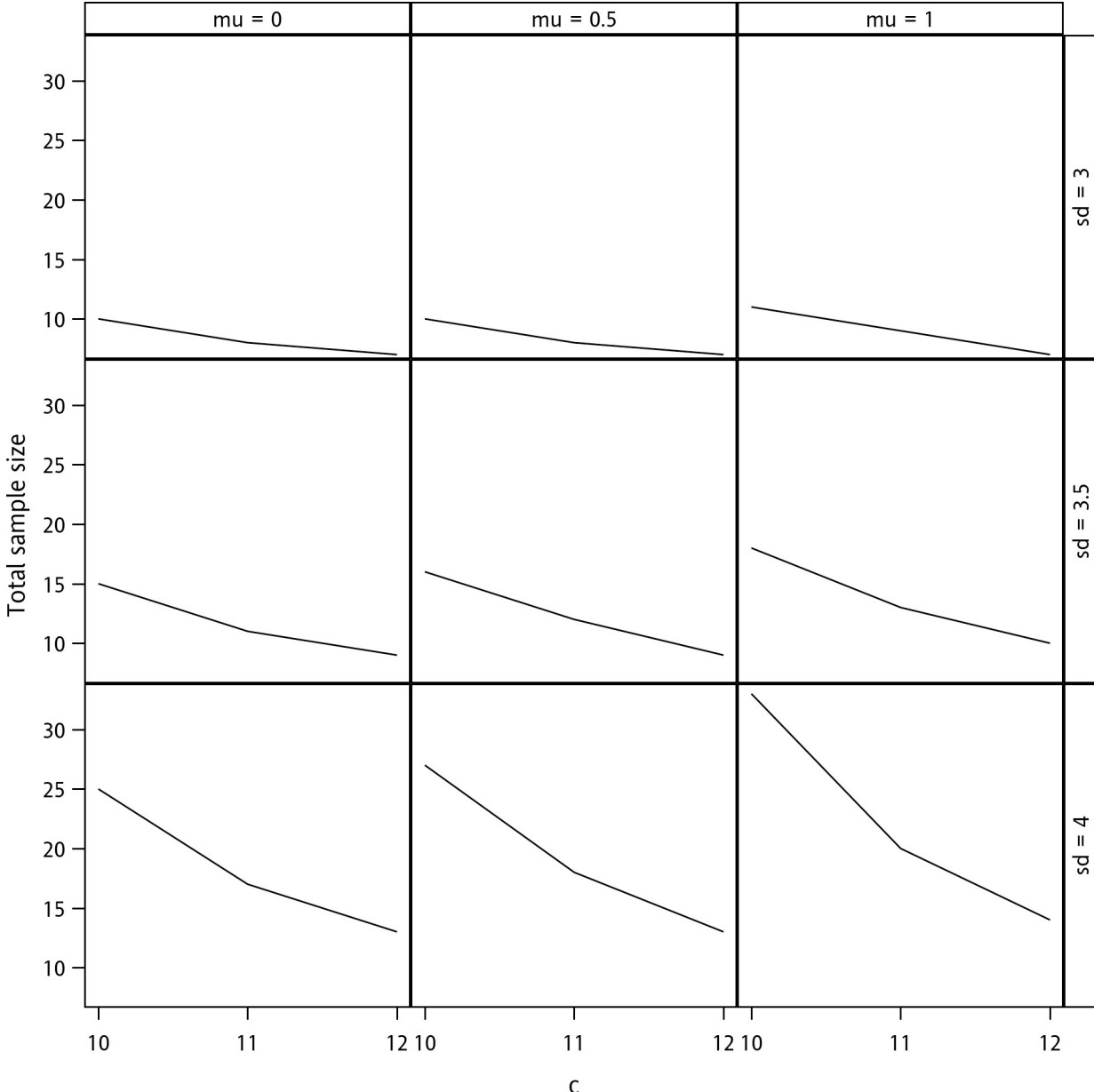

**Fig 1. Sample size determination with a confidence level of 90%, a content level of 90%, and a desired power of 80%.** The terms "mu" and "sd" denote $\mu$ and $\sigma$ respectively.

Though, we do not test the magnitude of the proportion $\gamma$. How large it is required for the proposed test is still of interest. Intuitively, a higher $\gamma$ leads to a wider interval. However, the width seems to be unimportant when applying a tolerance interval as a test statistic since we can always set an appropriate acceptance interval for the test. On the other hand, under normal assumptions, an increase in $\gamma$ results in the precision becoming more important in the assessment. Therefore, the issue may lie in how to balance the importance between accuracy and precision in our assessment.

Currently, the calculation of the exact tolerance factor is not prohibitive. For example, the k.factor() function in the tolerance package [10] for R calculates the exact k-factor. However, it

is known that the tolerance factor is a function of the sample size, while the required sample size is unknown for achieving a desired power and must be evaluated by the proposed sample size determination formula. Hence, an approximation of the tolerance factor with a closed-form can simplify the calculation. There are several approximations for the tolerance factor; for example, Krishnamoorthy and Mathew [7] suggested the use of the squared root of $(n-1)\chi^2_{n-1,\gamma,1/n}/\chi^2_{n-1,1-\alpha}$, where $\chi^2_{n-1,\gamma,1/n}$ is the $100\gamma$th quantile of a noncentral chi-square distribution with a noncentral parameter of $1/n$. Via an additional simulation with the same settings, we find that this approximation would overestimate the coverage probability (by approximately 92%) and requires a slightly larger sample size (1 to 3) to achieve the desired power than Howe's approximation. On the other hand, although Howe's approximation underestimates the coverage probability, the difference between Howe's result and the desired coverage probability is small and can be omitted. As a result, we recommend using Howe's approximation in the tolerance interval testing.

For testing $H_0:\mu\leq k_L$ or $\mu\geq k_U$ versus $H_a:k_L<\mu<k_U$ with prespecified constants $k_L$ and $k_U$, we can separate the test into two one-sided tests, where each side controls the type I error rate of $\alpha$, and the overall type I error rate is still $\alpha$. This is true because it is impossible for $\mu$ to be smaller than $k_L$ and larger than $k_U$ simultaneously. In contrast, for the proposed tolerance interval test, both the lower and upper acceptance margins might be exceeded simultaneously because of a large variability. As pointed out in the introduction, if necessary, we know that the tolerance interval test has to divide into two one-sided tests for positive bias and negative bias, respectively, and one two-sided test for variability. If so, whether each of the two one-sided tests with a significance level of $\alpha$ controls the overall type I error rate of $\alpha$ is in question. On the other hand, a two-sided tolerance interval with $100(1-\alpha)$% confidence itself has naturally controlled the type I error rate of $\alpha$ for the test. Therefore, it is unnecessary to separate the main test into three tests.

We, in fact, do not investigate the control of the type I error rate in the simulation. Alternatively, the preservation of the coverage probability is verified in the simulation study. The reason is that there are three scenarios fitting the null hypothesis, and this leads to difficulty in designing and analysing the simulation study. Second, as mentioned previously, by using a tolerance interval with $100(1-\alpha)$% confidence as the test statistic, the type I error rate can naturally be controlled for its corresponding test. As a result, if the coverage probability is satisfied, then the type I error rate can be controlled.

## Supporting information

**S1 Appendix. Derivation of the asymptotic normality of the lengths.**
(DOCX)

**S2 Appendix. SAS code for sample size determination.**
(DOCX)

**S3 Appendix. SAS code for p-value calculation.**
(DOCX)

## Acknowledgments

This research is part of collaborative work with Mycenax Biotech Inc. (Zhunan, Taiwan). Thanks are due to two referees for their detailed, constructive and thoughtful comments and suggestions which we believe have led to a significant improvement to this paper.

## Author Contributions

**Conceptualization:** Chieh Chiang.

**Investigation:** Chieh Chiang.

**Methodology:** Chieh Chiang.

**Writing – original draft:** Chieh Chiang.

**Writing – review & editing:** Chin-Fu Hsiao.

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
