## [Decision Letter · Decision Letter 0]

25 Nov 2020

PONE-D-20-31383

A tolerance interval testing for assessing accuracy and precision simultaneously

PLOS ONE

Dear Dr. Chiang,

Thank you for submitting your manuscript to PLOS ONE. After careful consideration, we feel that it has merit but does not fully meet PLOS ONE’s publication criteria as it currently stands. Therefore, we invite you to submit a revised version of the manuscript that addresses the points raised during the review process.

We look forward to receiving your revised manuscript.

Kind regards,

Alan D Hutson

Academic Editor

PLOS ONE

Journal Requirements:

2.We noticed you have some minor occurrence of overlapping text with the following previous publication, which needs to be addressed:

- https://onlinelibrary.wiley.com/doi/abs/10.1002/pst.2065

In your revision ensure you cite all your sources (including your own works), and quote or rephrase any duplicated text outside the methods section. Further consideration is dependent on these concerns being addressed.

Reviewers' comments:

Reviewer's Responses to Questions

**Comments to the Author**

1. Is the manuscript technically sound, and do the data support the conclusions?

Reviewer #1: Yes

Reviewer #2: Partly

2. Has the statistical analysis been performed appropriately and rigorously? 

Reviewer #1: Yes

Reviewer #2: No

3. Have the authors made all data underlying the findings in their manuscript fully available?

Reviewer #1: Yes

Reviewer #2: Yes

4. Is the manuscript presented in an intelligible fashion and written in standard English?

Reviewer #1: Yes

Reviewer #2: No

5. Review Comments to the Author

Reviewer #1: The authors explain in this article how to calculate the power when using a 2-sided tolerance interval (to assess jointly the mean and the standard deviation (accuracy and precision)) and the corresponding statistical hypotheses are clarified.

This paper is well written and concise. The power function is built under assymptotical property (bivariate normal distribution) which is somewhat disappointing (especially that the tolerance intervals are very useful for small sample sizes), but the results from the simulations show a very good agreement between this approximate power and the empirical power. As a consequence, the theory described in this paper can be implemented (sas code is given by the authors) and will bring an added-value to the scientific community.

Here are some remarks to improve the manuscript:

The authors should comment and explain what it is new in their paper compared to this one:

https://onlinelibrary.wiley.com/doi/epdf/10.1002/sim.8695

In p17 (pdf) - p11 article

"First, for the 27 combinations of the parameters, all of the empirical

powers are greater than the desired 80%, which demonstrates that the proposed

sample size determination can provide sufficient power for validating both accuracy

and precision simultaneously based on the two-sided tolerance interval."

This is somewhat confusing, power should be calculated for a given set of parameters. Bayesian power should be calculated if one wants to take into account the uncertainty around the parameters. The authors should elaborate or re-explain this clearly.

eq(6): define tau clearly

Dong et al. used a TOST approach with 1-sided intervals, but the 2-sided tolerance interval used in this paper is not new anyway. It is applied, for instance, in method comparison studies, and also in validation of analytical methods.

The approximate 2-sided tolerance interval by Howe has already been assessed in the statistical litterature (many papers, and book).

Reviewer #2: I understand the basic premise of seeking a general unifying inferential framework for tolerance intervals, but the flow of the paper is uneven and details are often vague. I think more intuitive discussions about the sample size determination would be beneficial, and overall a more expansive power study and analysis of said results would strengthen the work. More detailed comments are in the attachment.

6. PLOS authors have the option to publish the peer review history of their article (what does this mean?). If published, this will include your full peer review and any attached files.

Reviewer #1: **Yes: **Bernard G Francq

Reviewer #2: No

---

## [Author Response · Author response to Decision Letter 0]

2 Jan 2021

We sincerely thank the reviewers for their careful, constructive, thorough and thoughtful review which greatly improves the contents and presentation of our work. We have addressed all the comments in a point-to-point manner in the following and have incorporated all the referees’ suggestions into the revision.

---

## [Decision Letter · Decision Letter 1]

25 Jan 2021

Tolerance interval testing for assessing accuracy and precision simultaneously

PONE-D-20-31383R1

Dear Dr. Chiang,

We’re pleased to inform you that your manuscript has been judged scientifically suitable for publication and will be formally accepted for publication once it meets all outstanding technical requirements.

Kind regards,

Alan D Hutson

Academic Editor

PLOS ONE

Additional Editor Comments (optional):

Reviewers' comments:

Reviewer's Responses to Questions

**Comments to the Author**

1. If the authors have adequately addressed your comments raised in a previous round of review and you feel that this manuscript is now acceptable for publication, you may indicate that here to bypass the “Comments to the Author” section, enter your conflict of interest statement in the “Confidential to Editor” section, and submit your "Accept" recommendation.

Reviewer #1: All comments have been addressed

Reviewer #2: All comments have been addressed

2. Is the manuscript technically sound, and do the data support the conclusions?

Reviewer #1: Yes

Reviewer #2: Yes

3. Has the statistical analysis been performed appropriately and rigorously? 

Reviewer #1: Yes

Reviewer #2: Yes

4. Have the authors made all data underlying the findings in their manuscript fully available?

Reviewer #1: Yes

Reviewer #2: Yes

5. Is the manuscript presented in an intelligible fashion and written in standard English?

Reviewer #1: Yes

Reviewer #2: Yes

6. Review Comments to the Author

Reviewer #1: The major remarks from the reviewers have been addressed. This version of the manuscript is okay in my point of view.

Reviewer #2: The authors did a great job addressing both reviewers' comments. The paper is concise, and I think the paper reads much more clearly compared to the original submission. Good job!

7. PLOS authors have the option to publish the peer review history of their article (what does this mean?). If published, this will include your full peer review and any attached files.

Reviewer #1: **Yes: **Bernard G Francq

Reviewer #2: No

---

## [Editor Report · Acceptance letter]

27 Jan 2021

PONE-D-20-31383R1 

Tolerance interval testing for assessing accuracy and precision simultaneously 

Dear Dr. Chiang:

I'm pleased to inform you that your manuscript has been deemed suitable for publication in PLOS ONE. Congratulations! Your manuscript is now with our production department. 

Kind regards, 

on behalf of

Dr. Alan D Hutson 

Academic Editor

PLOS ONE